# The Role of «Novel» Biomarkers of Systemic Inflammation in the Development of Early Hospital Events after Aortic Valve Replacement in Patients with Aortic Stenosis

**DOI:** 10.3390/life13061395

**Published:** 2023-06-14

**Authors:** Vladimir Shvartz, Maria Sokolskaya, Artak Ispiryan, Madina Basieva, Polina Kazanova, Elena Shvartz, Sayali Talibova, Andrey Petrosyan, Teymuraz Kanametov, Sergey Donakanyan, Leo Bockeria, Elena Golukhova

**Affiliations:** 1Bakulev National Medical Research Center for Cardiovascular Surgery, Moscow 121552, Russia; 2National Medical Research Center for Therapy and Preventive Medicine, Moscow 101990, Russia

**Keywords:** aortic stenosis, systemic inflammation response index, systemic inflammation index, aggregate inflammation systemic index, neutrophil/lymphocyte ratio, platelet/lymphocyte ratio, monocyte/lymphocyte ratio

## Abstract

Introduction. The pathogenesis of aortic stenosis includes the processes of chronic inflammation, calcification, lipid metabolism disorders, and congenital structural changes. The goal of our study was to determine the predictive value of novel biomarkers of systemic inflammation and some hematological indices based on the numbers of leukocytes and their subtypes in the development of early hospital medical conditions after mechanical aortic valve replacement in patients with aortic stenosis. Materials and methods. This was a cohort study involving 363 patients who underwent surgical intervention for aortic valve pathology between 2014 and 2020. The following markers of systemic inflammation and hematological indices were studied: SIRI (Systemic Inflammation Response Index), SII (Systemic Inflammation Index), AISI (Aggregate Index of Systemic Inflammation), NLR (Neutrophil/Lymphocyte Ratio), PLR (Platelet/Lymphocyte Ratio), and MLR (Monocyte/Lymphocyte Ratio). Associations of the levels of these biomarkers and indices with the development of in-hospital death, acute kidney injury, postoperative atrial fibrillation, stroke/acute cerebrovascular accident, and bleeding were calculated. Results. According to an ROC analysis, an SIRI > 1.5 (*p* < 0.001), an SII > 718 (*p* = 0.002), an AISI > 593 (*p* < 0.001), an NLR > 2.48 (*p* < 0.001), a PLR > 132 (*p* = 0.004), and an MLR > 0.332 (*p* < 0.001) were statistically significantly associated with in-hospital death. Additionally, an SIRI > 1.5 (*p* < 0.001), an NLR > 2.8 (*p* < 0.001), and an MLR > 0.392 (*p* < 0.001) were associated with bleeding in the postoperative period. In a univariate logistic regression, SIRI, SII, AISI, and NLR were statistically significant independent factors associated with in-hospital death. In a multivariate logistic regression model, SIRI was the most powerful marker of systemic inflammation. Conclusion. SIRI, SII, AISI, and NLR as novel biomarkers of systemic inflammation were associated with in-hospital mortality. Of all markers and indices of systemic inflammation in our study, SIRI was the strongest predictor of a poor outcome in the multivariate regression model.

## 1. Introduction

Aortic valve stenosis is one of the most common cardiovascular diseases requiring surgical intervention. The incidence of this condition increases with age, and among patients over 65 years old, this condition is diagnosed in 2–5% of cases [1,2]. The pathogenesis of aortic stenosis involves processes of chronic inflammation, calcification, lipid metabolism disorders, and congenital structural changes [3].

Despite advances in medical science, along with the development of diagnostic and laboratory methods and pharmaceutical drugs, it is currently impossible to prevent the development and progression of this defect using modern medicamentous treatments. The most commonly performed surgery for aortic stenosis at present is aortic valve replacement in conditions of artificial blood circulation [4]. The mortality during isolated aortic valve repair does not exceed 3% [5]. In most cases, the risk of postoperative complications and death after surgery is associated with the preoperative comorbid status of the patient. Therefore, identifying predictors of unfavorable course development in the perioperative period is essential.

Preoperative and intraoperative risk factors have been well studied and defined. The former include age, reduced contractile capacity of the left ventricular myocardium, diagnosed diabetes mellitus, kidney and liver failure, arrhythmias, and stenotic coronary atherosclerosis. The latter comprise cardiopulmonary bypass time, aortic cross-clamp time, and the extent of the surgical intervention [6]. However, the search for other risk factors continues, because their early detection and correction can improve treatment outcomes.

To date, the prognostic significance of the so-called “novel” biomarkers of systemic inflammation in patients with various cardiovascular pathologies and chronic vascular diseases is actively studied [7,8,9]. These include the Systemic Inflammation Response Index (SIRI), the Systemic Inflammation Index (SII), the Aggregate Index of Systemic Inflammation (AISI), and various hematological indices based on the numbers of leukocytes and their subtypes (neutrophils, lymphocytes, and monocytes) [10,11,12]. However, there are few published studies which have investigated the possibility of using these biomarkers and indices in patients with acquired heart valve defects. Since these parameters of systemic inflammation are easily reproducible and potentially informative, we conducted this study. 

The objective of our study was to examine the predictive value of novel biomarkers of systemic inflammation and some hematological indices based on the numbers of leukocytes and their subtypes in the development of early hospital medical conditions after mechanical aortic valve replacement in patients with aortic stenosis.

## 2. Materials and Methods

### 2.1. Study Population

This study consists of a subanalysis of the database that includes all patients who underwent surgical intervention for aortic valve pathology in the cardiac surgery department at Bakulev Scientific Center of Cardiovascular Surgery. The general database encompasses patients with surgical aortic valve replacement (SAVR), mechanical and biological prostheses, transcatheter aortic valve replacement (TAVR), reconstructive aortic valve sparing operations, etc. The causes of aortic valve pathology in patients included in the general database are both congenital anomalies (bicuspid aortic valve) and acquired disorders (defects of rheumatic etiology, degenerative valve disease, infective endocarditis, etc.).

This article includes an analysis of the hospital stay (2014–2020) in patients who underwent aortic valve replacement with a mechanical prosthesis. Hemodynamically, these were patients with predominant aortic valve stenosis. Previously, we identified the clinical predictors of mortality in the early postoperative period in this cohort of patients [13].

For this analysis, the following pathologies were excluded from the database: concomitant pathologies of the mitral and tricuspid valves, ischemic heart disease, any arrhythmias including a history of atrial fibrillation, previous stroke, and any oncological disease including benign tumors. The patients were from the same department, and all of them underwent similar surgeries (with retrograde cardioplegia). 

### 2.2. Data Collection

Standard examination before surgery included the collection of clinical and anamnestic data, along with laboratory and instrumental tests. Confirmation of accompanying comorbid pathology was carried out by a physician specialized in the particular field (for example, an endocrinologist for diabetes, a pulmonologist for COPD, etc.). Confirmation or exclusion of ischemic heart disease was performed in all patients over 40 years old using selective coronary angiography. Transthoracic echocardiography was performed on a daily basis in all patients prior to the operation and in the early postoperative period in order to assess hemodynamics, the presence of pericardial effusion, myocardial contractile function, etc. Laboratory diagnostics included evaluation of blood parameters before the operation, as well as daily for 5 days after the operation and additionally on the day of discharge from the clinic (usually 7–8 days after the operation). ECG monitoring in the clinic was carried out in the first few days after the surgery using bedside monitors, and then using long-term ECG Holter monitoring until discharge. Intraoperative data, intensive care unit (ICU) data, and early postoperative period data were collected from the general electronic database of the MedWork clinic in compliance with all legal requirements.

Our research protocol conformed to the ethical recommendations of the Helsinki Declaration of 1975 and the Ethical Recommendations of the Government of the Russian Federation on Conducting Epidemiological Studies. This study was approved by the Local Ethics Committee of the Bakulev Center for Cardiovascular Surgery. Informed written consent from each patient before each procedure was obtained.

### 2.3. Definitions

Indices of systemic inflammation were calculated using the following formulas: SIRI (Systemic Inflammation Response Index) = neutrophil count × monocyte count ÷ lymphocyte count; SII (Systemic Inflammation Index) = neutrophil count × platelet count ÷ lymphocyte count; AISI (Aggregate Index of Systemic Inflammation) = neutrophil count × monocyte count × platelet count ÷ lymphocyte count; NLR (Neutrophil/Lymphocyte Ratio) = neutrophil count ÷ lymphocyte count; PLR (Platelet/Lymphocyte Ratio) = platelet count ÷ lymphocyte count; and MLR (Monocyte/Lymphocyte Ratio) = monocyte count ÷ lymphocyte count.

Development of acute kidney injury (AKI) was diagnosed based on KDIGO criteria (Kidney Disease Improving Global Outcomes) either given an increase in the level of blood serum creatinine of ≥0.3 mg/dL (≥26.5 mmol/L) within 48 h, or given its increase of ≥1.5-fold over the baseline value within 7 days. Postoperative atrial fibrillation (POAF) development was identified via both long-term ECG Holter monitoring and standard 12-lead ECG to determine the absence of visible regular P waves, the emergence of F waves, or irregular RR intervals on an ECG for over 30 s in the postoperative period.

A stroke was diagnosed on the basis of clinical data, the results of a CT scan of the brain, or an entry in the anamnesis made by a neurologist about a complication that took place. Bleeding was considered as an acute blood loss in the early postoperative period if a patient required infusion of blood components (plasma or red blood cells).

### 2.4. Endpoints

The primary endpoint was in-hospital mortality. Secondary endpoints were the development of the following non-fatal events: AKI, POAF, stroke/acute cerebrovascular accident (ACA), and bleeding. 

### 2.5. Surgery

Aortic valve replacement with a mechanical prosthesis was performed according to the standard procedure used in our department. All operations were performed in a planned manner in conditions of cardiopulmonary bypass, hypothermia, and pharmacological cold cardioplegia. Cardioplegia was performed retrogradely through the coronary sinus in all cases. A median sternum incision was used as an access to the heart. Combinations of aortic valve replacement with CABG, mitral valve interventions, and other manipulations were not considered in this article [14].

### 2.6. Statistical Analysis

The database included quantitative and qualitative variables. Quantitative data were tested for the normality of the distribution using the Shapiro–Wilk criterion. However, the distribution was always different from normal. To determine the effective diagnostic threshold values of the studied indices of systemic inflammation, we used an ROC analysis with curve construction and estimation of the area under the curve (AUC). The relationship between hospital mortality and the studied risk factors was assessed using single-factor and multivariate logistic regression analyses and represented by the odds ratio (OR), 95% confidence interval (CI), and *p*-value.

The data are presented in the form of median (Me) and interquartile range (Q1; Q3) when describing quantitative parameters, as well as in the form of an absolute number (n) and the fractional value (%) when describing qualitative parameters. In this analysis, we used the following software: Microsoft Office Excel and MedCalc (MedCalc Software Ltd., Ostend, Belgium).

## 3. Results

Our study encompassed 363 patients: 221 men (61%) and 142 women (39%). Their median age was 58.5 (44; 66.7) years. Table 1 presents the baseline clinical, initial instrumental, and initial laboratory characteristics of patients and their drug therapy before surgery. The operative data of patients and hospital outcomes of patients are presented in Table 2.

An ROC analysis was performed to determine threshold values for systemic inflammation indices associated with the primary and secondary endpoints of the study (Table 3 and Figure 1).

When performing the univariate and multivariate logistic regression analyses, we also added clinical and operational predictors of a lethal outcome that we found earlier in this cohort of patients [13] (Table 4). 

## 4. Discussion

This study included solely patients with isolated aortic stenosis in accordance with our inclusion/exclusion criteria. We initially limited the potential impact of other pathologies by excluding patients with concomitant pathology of other heart valves, ischemic heart disease, arrhythmias, and cancer from the analysis. Consequently, as can be seen from Table 1, the severity of concomitant pathology was insignificant. The frequency of the primary endpoint (in-hospital mortality) was 2% (seven patients). The frequencies of secondary endpoints in this study were as follows: 13% for AKI, 10% for POAF, 2% for stroke/ACA, and 0.8% for bleeding. 

In the course of the ROC analysis, statistically significant threshold values associated with the development of in-hospital mortality were obtained for all systemic inflammation indices. Statistically significant levels were also obtained for some of the studied indices associated with the development of bleeding in the early postoperative period. No statistically significant levels were observed for inflammation indices associated with the development of AKI, POAF, and stroke/ACA.

The univariate analysis revealed that novel biomarkers of systemic inflammation and NLR were statistically significant independent factors associated with in-hospital mortality. Some of the previously obtained operational data and the level of hemoglobin before the surgery had prognostic values as well. In the multivariable logistic regression analysis, only two factors were statistically significant (SIRI and ACC time), while all other factors were leveled off by the multifactorial nature of the model.

A number of studies have demonstrated that the level of neutrophils, lymphocytes, and platelets plays an important role in the development of chronic inflammation and the progression of cardiovascular diseases. In some studies, biomarkers such as C-reactive protein, NLR, and PLR demonstrated their prognostic values for the risk of postoperative complications and mortality [15,16,17]. For example, in the study by Russu E. et al., the preoperative role of NLR and PLR in the medium-term outcome of patients surgically revascularized for femoropopliteal disease was studied. It was shown that a high value of preoperative NLR and PLR determined at hospital admission was strongly predictive of primary patency failure (12 months after revascularization). Additionally, elevated ratio values are an independent predictor for a higher amputation rate [18].

In recent years, the possibility of using the SII as a predictor of unfavorable outcomes among patients with acquired heart defects was actively investigated. This marker of systemic inflammation takes into account the level of neutrophils, platelets, and lymphocytes, thereby reflecting both inflammatory and immunological activation, as well as activation of the coagulation cascade [17]. The SII was studied in patients with chronic heart failure, neoplasms, and nervous system disorders [7,8]. In addition, this index was found to be statistically significant for predicting adverse outcomes in scheduled myocardial revascularization, and also for predicting the frequency of POAF and other complications in the postoperative period in cardiac surgery patients [19,20,21]. Tosu et al. conducted a follow-up of 120 patients undergoing transcatheter aortic valve implantation (TAVI) for aortic stenosis, in whom the SII was an independent predictor of postoperative complications [22]. The study by Xiang J et al. analyzed a cohort of patients (n = 431) with various valve pathologies who underwent surgical intervention. It was established that a high level of preoperative systemic inflammation, measured with the SII, was statistically significantly associated with an increased risk of postoperative complications, the 30 day postoperative mortality, the duration of stay in the ICU, and the duration of the hospital stay [16].

Another marker of inflammation, the SIRI, combines neutrophils, monocytes, and lymphocytes in peripheral blood and is an indicator of low-grade chronic inflammation. Its elevated values are associated with the risk of developing a stroke, a complicated postoperative period, and a 5 year mortality rate after off-pump CABG surgery [23,24]. In our study, this parameter exhibited the strongest correlation with the primary endpoint. 

The AISI is less studied and less common in the literature. It takes into account the ratio of all subtypes of leukocytes and platelets. Study [25] presented data that could be used as a predictor of a complicated postoperative period for thoracic surgery. However, this study had a relatively small sample size (157 patients), which limited the reliability of the obtained results. Among the published studies, we did not find any in which the AISI was analyzed for cardiac surgery. Therefore, further investigation of this marker in cardiac surgery is required.

The NLR was proposed as a straightforward marker of inflammation that is useful for the long-term prognosis in ischemic heart disease and as a prognostic risk factor for the development of postoperative complications, POAF, and mortality after cardiovascular surgery [26,27,28,29,30]. Neutrophils are participants of the inflammatory response. They can secrete inflammatory mediators and are involved in the processes of phagocytosis and chemotaxis. Activated neutrophils release genetic material, thereby forming chromatin networks known as neutrophil extracellular traps (NETs). The latter induce oxidative stress and dysfunction of endothelial cells, along with contributing to the accumulation of prothrombotic molecules [9,31,32]. These highly sensitive markers are difficult to identify in routine clinical practice, whereas an increase in the number of neutrophils in peripheral blood can be considered as a basic method for monitoring neutrophil activation [16]. Including the NLR in a prognostic model, along with clinical preoperative factors and echocardiographic parameters, can help identify a group of patients requiring more careful postoperative monitoring due to an increased risk of complications. In our study, the NLR was statistically significantly associated with in-hospital mortality in a univariate model. 

Monocytes play an important role in the formation of atherosclerotic plaque. The interaction between monocytes and endothelial cells leads to a local imbalance between the processes of damage and repair, which negatively affects the stability of the plaque. In the literature, the MLR is considered as a risk factor for venous bypass patency after revascularization, as well as a marker of a poor 5 year prognosis after off-pump CABG [33,34,35]. Endothelial dysfunction plays an important role in the pathogenesis of aortic stenosis; thus, we examined the MLR in this cohort of patients. Although the ROC analysis identified diagnostic levels of MLR that were statistically significantly associated with in-hospital mortality and early bleeding, we did not obtain statistically significant data on this indicator in the regression analysis.

The PLR is considered as a prognostic risk factor for cardiovascular diseases and a marker of perioperative complications. Additionally, some studies established a correlation between the PLR and the severity of coronary artery lesions [36,37,38,39]. Platelets are formed from the mononuclear phagocyte system. They interact with leukocytes and endothelial cells of blood vessels; activate and induce adhesion and transport of monocytes; participate in the release of interleukin 1 (IL-1), tumor necrosis factor, and other factors of inflammation; and are involved in the processes of local myocarditis and myocardial fibrosis in patients with valve pathology [40]. An elevated level of platelets is associated with a poor prognosis for patients with cardiovascular disease [41]. Platelets participate in the onset of the atherosclerotic process and can maintain vascular inflammation. Additionally, they attract immune cells (monocytes and neutrophils), acting as mediators of tissue hemostasis, which can modulate the microenvironment of the atherosclerotic plaque [36,42,43]. Similar to the MLR, the PLR level in our study was statistically significant in the ROC analysis, but did not yield significance in the logistic regression.

A feature of the clinical picture of aortic stenosis is a long-term asymptomatic course of the defect. Therefore, many patients develop chronic heart failure, which triggers the processes of chronic inflammation, oxidative stress, and neurohormonal activation, leading to an increase in the level of cortisol in the blood plasma and the release of catecholamines. This suppresses lymphocyte differentiation and proliferation, followed by increased lymphocyte apoptosis [44]. Consequently, a reduced lymphocyte count is an independent predictor of poor survival in patients with chronic and progressive heart failure [45].

Currently, aortic valve replacement under the condition of a cardiopulmonary bypass (CPB) is the most commonly performed operation in this group of patients. Surgery under a CPB is a triggering factor for the development of an inflammatory response and increases the risk of complications. When choosing the strategy for surgical treatment of ischemic heart disease via open-heart surgery, preference is given to the off-pump technique, due to the undesirable side effects of a CPB during the on-pump technique [46]. For example, 70–90% of patients who underwent on-pump CABG were diagnosed with systemic inflammation [47]. The activation of a CPB triggers inflammatory processes and disrupts the endothelial function due to hemodilution, endothelial damage, hypothermia, nonpulsatile blood flow, administration of heparin, and the release of proinflammatory mediators. The inflammatory response associated with neutrophils can cause the activation of endothelial cells. In turn, endothelial cytotoxicity activates intracellular mechanisms for the production of nitric oxide. During a CPB, when an inflammatory response is generated, neutrophils are attracted to the site of tissue damage [48]. With a complex cascade of mediators, including tumor necrosis factor (TNF-α) and IL-1, endothelial and parenchyma cells are stimulated to release chemokines that attract neutrophils [49]. Adhesion and transmigration of neutrophils ultimately leads to transmigration into the extravascular space. Published sources have described systemic complications of a CPB such as platelet dysfunction, activation of the coagulation cascade, and production of free radicals [50].

Tissue damage involving neutrophils also occurs during reperfusion. Platelets and neutrophils may play a role in the so-called no reflow phenomenon, when they become trapped inside capillaries, causing hypoperfusion of hypoxic tissues [50,51]. Neutrophils are activated not only in the presence of a CPB. Contact of blood with a foreign surface triggers the activation of factor XII (FXII), with the formation of FXIIa and FXIIf [52]. Contact of the foreign body surface with blood components causes an inflammatory response. FXIIf is responsible for the activation of the complement system and complement membrane attack, lysis, and cell death. Endothelial cells produce procoagulant and anticoagulant factors that maintain hemorheology. Contoured artificial surfaces lead to the activation of coagulation cascades. Furthermore, thrombin and coagulation factor Xa, which are abundantly produced during CPB, activate platelets through their thrombin receptors of the PAR family and leukocytes [53]. In turn, activated platelets release neutrophil activators such as interleukins 6 and 8 (IL-6 and IL-8) [54]. Hence, coagulation is associated with the process of inflammation. 

The inflammatory cell count in peripheral blood is used in clinical practice to predict the course of the postoperative period. Assessment of such indicators can improve the results of surgical treatment via better identification of patients with a higher risk of a poor outcome. Of course, further research is needed to establish the role of inflammation markers in specific groups of patients, taking into account the presence of diabetes mellitus, chronic kidney disease, and peripheral arterial disease.

### Limitations of the Study

This analysis had disadvantages associated with the retrospective approach. Firstly, since all the data were collected from the general electronic database of our clinic MedWork with standard data entry, this does not exclude partial data loss. Secondly, a small number of patients were included in the analysis. However, this is explained by very strict inclusion and exclusion criteria. We excluded concomitant diseases, making the group more homogeneous in order to focus only on the effect of markers of systemic inflammation on the studied outcomes.

## 5. Conclusions

Elevated levels of novel biomarkers of systemic inflammation (SIRI, SII, and AISI), along with the level of the NLR, were associated with in-hospital mortality. Of all markers and indices of systemic inflammation in our study, the SIRI and CPB time were the strongest predictors of a poor clinical outcome in the multivariate regression model. 

The possibility of the routine use of systemic inflammation indices for patients with aortic stenosis as predictors of the complications in the perioperative period is quite promising.

## Figures and Tables

**Figure 1 life-13-01395-f001:**
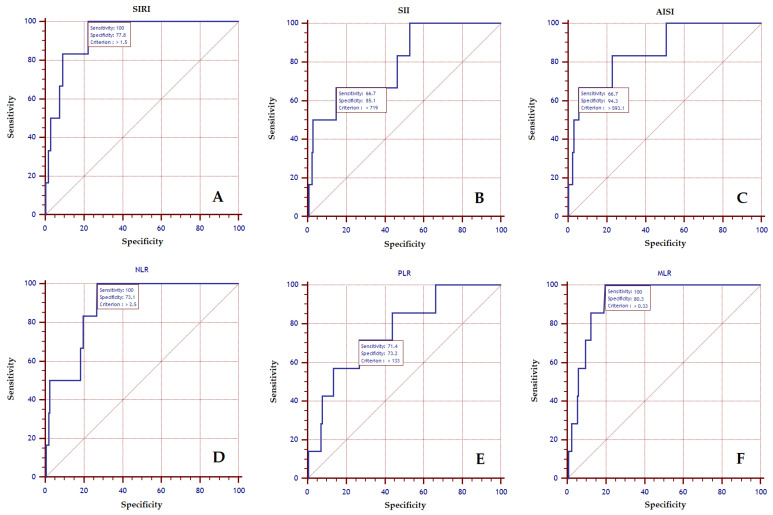
ROC curves of systemic inflammation indices for lethal outcome. (**A**) SIRI, (**B**) SII, (**C**) AISI, (**D**) NLR, (**E**) PLR, (**F**) MLR.

**Table 1 life-13-01395-t001:** Clinical, instrumental, and laboratory characteristics of patients and their drug therapy before surgery.

Parameters	Value (n = 363)
Age, years	58.5 (44; 66.7)
Male, %	221 (61)
BMI, kg/m^2^	27.8 (24.5; 30.9)
BSA, m^2^	0.2 (0.19; 0.21)
Hypertension, %	134 (37)
Smoking, n (%)	40 (11)
CHF NYHA class III-IV, n (%)	134 (37)
Diabetes, n (%)	25 (7)
COPD, n (%)	7 (2)
CKD, n (%)	14 (4)
Initial instrumental data	
LVEF, %	63 (58; 66)
iEDD	2.6 (2.3; 3.0)
iESD	1.7 (1.5; 1.9)
iEDV	67.9 (51.7; 89.4)
iESV	23 (18; 33)
Peak gradient, mm Hg	90 (56; 107)
Mean gradient, mm Hg	52 (42; 67)
Fibrous ring of the aortic valve, mm	24 (22; 26)
LA volume, mL^3^	104 (80; 133)
Initial laboratory data	
Hemoglobin, g/L	139 (128; 148)
Hematocrit, %	42 (39; 44)
WBC, 10^9^/L	7.1 (5.9; 8.4)
Neutrophils, 10^9^/L	4.4 (3.6; 5.5)
Neutrophils, %	60 (53; 65)
Glucose, mmol/L	5.2 (4.8; 5.6)
Fibrinogen, g/L	4.0 (3.5; 4.5)
Creatinine, mkmol/L	80 (70; 94)
eGFR mL/min	95 (77; 114)
eGFR mL/min per 1.73 m^2^ (MDRD)	81 (71; 97)
Initial drug therapy	
Beta-blockers, %	41
ACE inhibitors, %	29
ARA, %	11
Calcium antagonists, %	9
Statins, %	20
Nitrates, %	3
Thiazide diuretics, %	9
Loop diuretics, %	12
Potassium-sparing diuretics, %	15

BMI—body mass index; BSA—body surface area; CHF NYHA—chronic heart failure New York Heart Association classification; COPD—chronic obstructive pulmonary disease; CKD—chronic kidney disease; LVEF—left ventricular ejection fraction; iEDD—index end diastolic diameter; iEDV—index end diastolic volume; iESD—index end systolic diameter; iESV—index end systolic volume; LA—left atrium; WBC—white blood cells; eGFR—estimated glomerular filtration rate; MDRD—modification of diet in renal disease; ACE—angiotensin-converting enzyme; ARA—angiotensin receptor antagonist.

**Table 2 life-13-01395-t002:** Operative data of patients and hospital outcomes.

Parameters	Value (n = 363)
Operative data	
CPB time, min	130 (113; 153)
ACC time, min	65 (58; 76)
ICU time > 2 days, n (%)	42 (11.6)
Clinical outcomes and complications	
Mortality, n (%)	7 (2)
AKI, n (%)	47 (13)
POAF, n (%)	36 (10)
Stroke/ACA, n (%)	7 (2)
Bleeding, n (%)	3 (0.8)
Length of stay, days	7 (6; 8)

CPB—cardiopulmonary bypass; ACC—aortic cross-clamp; ICU—intensive care unit; AKI—acute kidney injury; POAF—postoperative atrial fibrillation; ACA—acute cerebrovascular accident.

**Table 3 life-13-01395-t003:** Results of ROC analyses for quantitative values of systemic inflammation indices associated with the development of complications.

Parameters	Cut-Off Point	AUC (CI)	Se	Sp	*p*
Death					
SIRI	>1.5	0.927 (0.891–0.954)	100	77.8	<0.001 *
SII	>718	0.800 (0.750–0.844)	66.7	85.1	0.002 *
AISI	>593	0.857 (0.812–0.896)	66.7	94.3	<0.001 *
NLR	>2.48	0.884 (0.841–0.918)	100	73.1	<0.001 *
PLR	>132	0.763 (0.716–0.806)	71.4	73.2	0.004 *
MLR	>0.332	0.920 (0.886–0.947)	100	80.3	<0.001 *
Acute kidney injury					
SIRI	>0.898	0.549 (0.483–0.614)	72.7	46.3	0.073
SII	>321	0.547 (0.481–0.611)	87.9	26	0.379
AISI	>243	0.568 (0.501–0.632)	63.6	56.5	0.202
NLR	>1.77	0.532 (0.467–0.597)	72.7	41.5	0.535
PLR	<92.6	0.489 (0.430–0.547)	39.5	69.8	0.833
MLR	<0.133	0.498 (0.437–0.558)	13.5	95.4	0.966
POAF					
SIRI	>1.2	0.532 (0.473–0.591)	48.3	60.9	0.574
SII	>567	0.534 (0.475–0.592)	44.8	69.5	0.589
AISI	>388	0.517 (0.457–0.575)	31	80.8	0.784
NLR	>1.9	0.563 (0.504–0.620)	72.4	44.9	0.281
PLR	>110	0.514 (0.461–0.567)	57.9	54	0.794
MLR	>0.182	0.513 (0.459–0.567)	86.5	23.3	0.787
Stroke or ACA					
SIRI	>1.45	0.600 (0.513–0.682)	66.7	72.6	0.685
SII	>553	0.556 (0.470–0.639)	66.7	67.4	0.774
AISI	>337	0.593 (0.506–0.675)	66.7	70.4	0.668
NLR	>2.43	0.531 (0.446–0.616)	66.7	66.7	0.892
PLR	<113	0.629 (0.556–0.698)	100	44.9	0.308
MLR	>0.3	0.521 (0.445–0.597)	66.7	68.6	0.921
Bleeding					
SIRI	>1.5	0.843 (0.771–0.900)	100	75	<0.001 *
SII	>428	0.500 (0.414–0.586)	100	46.7	0.999
AISI	>231	0.616 (0.528–0.698)	100	48.5	0.391
NLR	>2.8	0.766 (0.687–0.834)	100	75.9	<0.001 *
PLR	<112	0.567 (0.493–0.639)	100	46.2	0.540
MLR	>0.392	0.904 (0.849–0.943)	100	88.9	<0.001 *

AUC—area under curve; CI—confidence interval; Se—sensitivity; Sp—specificity; SIRI—systemic inflammation response index; SII—systemic inflammation index; AISI—aggregate inflammation systemic index; NLR—neutrophils lymphocytes ratio; PLR—platelets lymphocytes ratio; MLR—monocytes lymphocytes ratio; POAF—postoperative atrial fibrillation; ACA—acute cerebrovascular accident; *—statistically significant differences.

**Table 4 life-13-01395-t004:** Uni- and multi-variate logistic regression analyses for a lethal outcome.

Parameters	Univariate Logistic Regression Analysis OR (95% CI)	*p*	Multivariate Logistic Regression Analysis OR (95% CI)	*p*
SIRI	1.143 (1.014–1.289)	0.029 *	2.533 (1.001–6.414)	0.049 *
SII	1.003 (1.001–1006)	0.015 *	-	-
AISI	1.000 (1.000–1.001)	0.042 *	-	-
NLR	1.110 (1.025–1.203)	0.011 *	-	-
PLR	1.001 (1.000–1.002)	0.068	-	-
MLR	1.736 (0.980–3.073)	0.058	-	-
BMI	-	-	-	-
Hemoglobin, g/L	0.957 (0.921–0.995)	0.041 *	-	-
Hematocrit, %	0.895 (0.795–1.001)	0.091	-	-
CPB time, min	1.021 (1.007–1.037)	<0.001 *	-	-
ACC time, min	1.048 (1.029–1.075)	<0.001 *	1.046 (1.019–1.073)	<0.001 *
Diabetes	-	-	-	-

OR—odds ratio; CI—confidence interval; SIRI—systemic inflammation response index; SII—systemic inflammation index; AISI—aggregate inflammation systemic index; NLR—neutrophils lymphocytes ratio; PLR—platelets lymphocytes ratio; MLR—monocytes lymphocytes ratio; BMI—body mass index; CPB—cardiopulmonary bypass; ACC—aortic cross-clamp; *—statistically significant differences.

## Data Availability

The primary data analyzed in this study are not publicly available due to the policy of access to clinical data of the Bakulev Center for Cardiovascular Surgery of the Ministry of Health of the Russian Federation. However, some parameters that do not contain personal information can be provided by the corresponding author upon reasonable request.

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
