# Peer review of "The Role of «Novel» Biomarkers of Systemic Inflammation in the Development of Early Hospital Events after Aortic Valve Replacement in Patients with Aortic Stenosis"

_life, 2023, doi:10.3390/life13061395_

Round 1

Reviewer 1 Report

Dear editor,

I read with interest the manuscript written by Shvartz et al., regarding the the predictive value of novel biomarkers of systemic inflammation and some hematological indices based on the numbers of leukocytes and their subtypes in the development of early hospital medical conditions after the mechanical aortic valve replacement in patients with aortic stenosis. The authors result show that elevated levels of systemic inflammation novel biomarkers (SIRI, SII, AISI), along with the level of NLR, were associated with in-hospital mortality. In addition, elevated levels of SIRI, NLR and MLR were associated with early postoperative bleeding. Of all markers and indices of systemic inflammation in our study, SIRI and CPB time were the strongest predictors of the poor clinical outcome in the multivariate regression model

Overall, the manuscript is well written and very well structured, and easy to read. I want to congratulate the authors for their work.

However, I have some suggestion to improve the quality of the paper.

1.     In Introduction, the authors can improve this section, regarding the regarding the predictive value of hematological ratios and vascular pathology. See the following articles:

- https://doi.org/10.3390/life12060822

- https://doi.org/10.3390/medicina58101502       

- https://doi.org/10.3390/life12091447

2.  In Discussion section, I suggest improving this part by comparing the optimal cut-off value and predictive role of the hematological markers in patients with other chronic vascular disease such Peripheral Arterial Disease, or Carotid Disease. There will be very interesting to see the difference. See the following articles:

-https://doi.org/10.3390/ijerph192113934  

-https://doi.org/10.3390/jcm11092620

Author Response

We are grateful to the Reviewer for the high evaluation of our article.

  1. We have expanded the introduction section by adding data on the prognostic value of hematological indicators and vascular pathology, taking into account the recommended references.
  2. We are grateful to the Reviewer for the high evaluation of our article. We have expanded the introduction section by adding data on the prognostic value of hematological indicators and vascular pathology, taking into account the recommended references.

Reviewer 2 Report

The authors report an examination of the value of calculated indices upon mortality and other secondary outcomes after aortic valve replacement.  The study does not follow well-established principles for reporting the ADDITIONAL value of biomarkers upon clinical outcomes.

Did you include TAVR in this study or was it just in the primary cohort?  The Methods seem to imply that.  

I don't understand why the exclusion criteria were so broad.  Coronary artery disease, cancer, atrial fibrillation,concomitant mitral or tricuspid disease and non-hematological cancers are common in this age group and should be included to broaden applicability of the results.  Your sample size would then be much larger and the study would then be more broadly applicable.

The definition of bleeding is unsatisfactory.

In Table 2. you have ICU time > 2 days, n (%) 11.6, but there is no N in that column.

You say "Laboratory diagnostics included evaluation of blood parameters before the operation, as well as daily for 5 days after the operation and additionally on the day of discharge from the clinic (usually 7-8 days after the operation)".  So a few questions:

1.  Please state that the laboratory data in Table 2 is preoperative data.

2.  The results in Table 3 - were they from preoperative SIRI, SII etc data?

3.  Please tell us how you estimated the cutoff points.

You say "When calculating the univariate and multivariate logistic regression analyses, we also added clinical and operational predictors of a lethal outcome that we found earlier in this cohort of patients [10] (table 4).", 

1.  Please add these variables to the model rep[ort in Table 4.  The very few variables you currently have in Table 4 are of little importance

2.  Please tell us the ADDITIONAL value of adding your hematologic variables to a base clinical model.

3.  AISI is not significant in the multivariate model.  Please delete it from the model.

4.  It looks like AISI has marginal univariate significance, and certainly no importance, based on the data reported in Table 4.

5.  Your prior paper (ref 10) has frequent ventricular ectopy as a predictive variable, but it is not included in this paper. There is no apparent accounting for age in the model.  In every model of mortality I have seen it is one of the top predictors.  

6.  Only reporting the univariate predictors for the secondary variables is inadequate.  If there is no additional predictive value of SIRI and other predictors in multivariate models, please overtly say so.  In the abstract you say: "Also, levels of SIRI>1.5 (Ñ€<0.001), NLR>2.8 (Ñ€<0.001) and MLR >0.392 (Ñ€<0.001) were associated with bleeding in the postoperative period."  This not a realistic finding unless you have a multivariable model that shows additional value to the predictors.  

7.  I suspect your previously reported (ref 10) clinical model is weak. Calculating the STS mortality index for each patient and including it in a model would provide much greater importance to results you generate. 

With only 7 deaths you should be vastly underpowered to examine the primary outcome. The same is probably true for the secondary outcomes.  There is no a priori estimation of either power or required sample size for this analysis.  I am very surprised you observed significant results from this analysis.

There some interesting errors of English idiom and grammar in the paper.

Author Response

We are grateful to the Reviewer for evaluating our article.

Did you include TAVR in this study or was it just in the primary cohort?  The Methods seem to imply that.  

It is described in Section 2.1. We did not include TAVR in the analysis. "This article includes an analysis of the hospital stay (2014-2020) of patients who underwent aortic valve replacement with a mechanical prosthesis."

I don't understand why the exclusion criteria were so broad.  Coronary artery disease, cancer, atrial fibrillation,concomitant mitral or tricuspid disease and non-hematological cancers are common in this age group and should be included to broaden applicability of the results.  Your sample size would then be much larger and the study would then be more broadly applicable.

We are grateful to the Reviewer for this comment. All the listed diseases (coronary heart disease, cancer, AF) also have a systemic inflammatory component in their pathogenesis, on the one hand, and are independent predictors of the outcomes studied by us, on the other hand. For example, preoperative AF increases the risk of developing of postoperative AF tenfold, thus it may affect on the results of the study of the systemic inflammation markers role (SIRI, SII, AISI, etc.) in the development of AF. Combined aortic valve replacement and CABG surgery increase the risks of mortality and bleeding. Therefore, we excluded these diseases, making the group more homogeneous in order to focus only on the effect of markers of systemic inflammation on the studied outcomes. Of course, the sample size would be larger, but there would increase the number of influencing concomitant factors.

The definition of bleeding is unsatisfactory.

We are grateful to the Reviewer for this comment. We have chosen such a criterion from a purely practical side. The protocol and indications for blood components infusion in patients with acute blood loss are clearly described in our clinic. Therefore, the fact of using blood components in the early postoperative period in patients with acute blood loss was considered by us an important outcome in our practice.

In Table 2. you have ICU time > 2 days, n (%) 11.6, but there is no N in that column.

We are grateful to the Reviewer for noticing this. This is our typo. We have added this information.

You say "Laboratory diagnostics included evaluation of blood parameters before the operation, as well as daily for 5 days after the operation and additionally on the day of discharge from the clinic (usually 7-8 days after the operation)".  So a few questions:

  1.  Please state that the laboratory data in Table 2 is preoperative data.
  2.  The results in Table 3 - were they from preoperative SIRI, SII etc data?
  3.  Please tell us how you estimated the cutoff points.

  1. Yes, the Table 2 shows the initial laboratory data before the operation. After all, we are testing the hypothesis: do these biomarkers reflect the initial status of chronic systemic inflammation in patients with aortic stenosis? And are they predictably significant? We have added this information.
  2. Yes, the Table 3 also shows the initial data. In the early postoperative period, the markers will change depending on the severity of the acute inflammatory response to the operation with artificial blood circulation. We were interested in the initial status.
  3. Interesting question from the Reviewer. If you mean the technical component, then the algorithm is represented by the following steps (using SIRI as an example):

The table contains quantitative parameters SIRI for each patient, as well as the logical state of the outcome, mortality, marked "1" if "yes" and "0" if "no".

In the MedCalc program (MedCalc Software Ltd., Belgium), we select "Statistics" - "ROC curves" - "ROC curve analysis" (see screenshot).

You say "When calculating the univariate and multivariate logistic regression analyses, we also added clinical and operational predictors of a lethal outcome that we found earlier in this cohort of patients [10] (table 4)." 

Please add these variables to the model rep[ort in Table 4.  The very few variables you currently have in Table 4 are of little importance

I didn't quite understand the reviewer's comment. Previously, we conducted an analysis of clinical predictors of mortality in this cohort of patients (https://doi.org/10.3390/pathophysiology29010010), where the following statistically significant factors were obtained: ACC time, CPB time, Hemoglobin, Hematocrit and others. Some of them were factors related to concomitant pathology (coronary heart disease, history of myocardial infarction, etc.) and combined interventions (MV repair, TV repair). But considering that in this work we study a more homogeneous group of patients with isolated aortic stenosis, we excluded other pathologies from the analysis, so we took into this model the parameters that we had previously obtained and were also available in this group of patients.

Please tell us the ADDITIONAL value of adding your hematologic variables to a base clinical model.

The additional value is due to the fact that the calculation of these inflammation markers (SIRI, SIRI, AISI, etc.) in peripheral blood is affordable, cheap in clinical practice and can be used in predicting the course of the postoperative period. Evaluation of such indicators can improve the results of surgical treatment by better identifying patients with a higher risk of a worse outcome.

AISI is not significant in the multivariate model.  Please delete it from the model.

Yes, indeed, AISI had no statistical significance in the multifactor model. It's marked. However, the program did not exclude it on its own, like other insignificant factors. However, if the Reviewer thinks so, we will remove AISI from the model.

It looks like AISI has marginal univariate significance, and certainly no importance, based on the data reported in Table 4.

This is not quite true. In the one-factor model, AISI had a statistical significance of p = 0.042*, although borderline.

Your prior paper (ref 10) has frequent ventricular ectopy as a predictive variable, but it is not included in this paper. There is no apparent accounting for age in the model.  In every model of mortality I have seen it is one of the top predictors.  

Yes, it really so, as the Reviewer writes. I’ve already said a little above that we studied patients with isolated aortic stenosis only in this analysis and we excluded other pathologies. Therefore, all initial arrhythmias were excluded, and frequent ventricular ectopia was also excluded. A smaller sample of patients in this analysis is also associated with it.

Only reporting the univariate predictors for the secondary variables is inadequate.  If there is no additional predictive value of SIRI and other predictors in multivariate models, please overtly say so.  In the abstract you say: "Also, levels of SIRI>1.5 (Ñ€<0.001), NLR>2.8 (Ñ€<0.001) and MLR >0.392 (Ñ€<0.001) were associated with bleeding in the postoperative period."  This not a realistic finding unless you have a multivariable model that shows additional value to the predictors.

I understood the Reviewers point of view, he is really right. To be honest, we didn't want to overload the article with additional models related to other outcomes. Therefore, we stopped only at the multifactor model associated with the primary endpoint. To avoid misunderstanding, we will remove this phrase from the article.

I suspect your previously reported (ref 10) clinical model is weak. Calculating the STS mortality index for each patient and including it in a model would provide much greater importance to results you generate. 

There is no reference index for calculating the risk of mortality. Yes, indeed STS score is calculated and validated on a large number of patients, it has a great statistical power. However, it is calculated on the American population and it cannot be used on our patients. In STS score, it is not even possible to choose a race suitable for our patients. Therefore, we study our sample and our experience.

That is the right of the Reviewer to express his opinion, but I do not take this into account as a remark and comment that something needs to be corrected. After all, it's just his opinion. Everyone has their own point of view.

With only 7 deaths you should be vastly underpowered to examine the primary outcome. The same is probably true for the secondary outcomes.  There is no a priori estimation of either power or required sample size for this analysis.  I am very surprised you observed significant results from this analysis.

We were interested to see whether these so-called "new" biomarkers of systemic inflammation, which are essentially calculated indicators from the leukocyte formula, are associated with hospital adverse outcomes. We have received interesting data that we want to share.

We are grateful to the Reviewer for evaluating our article!

Reviewer 3 Report

The authors evaluated several biomarkers of systemic inflammation in 363 patients undergoing mechanical aortic valve replacement. Several biomarkers were significantly associated with elevated in-hospital mortality, but SIRI was the strongest. There is already literature about the predictive value of inflammation markers available in TAVI patients, but it is very interesting that there is the same association in healthier, younger patients undergoing mechanical valve replacement.

Minor comment:

-        Were patients included consecutively? How many patients were excluded due to missing data?

-        A paragraph with limitations is missing. Special limitations of this analysis include missing long-term follow up, retrospective analysis.

-        What should clinicians do when inflammatory markers are elevated? Reschedule? Cancel?

Author Response

We are grateful to the Reviewer for the high evaluation of our article.

-        Were patients included consecutively? How many patients were excluded due to missing data?

Section 2.1. describes in detail how we selected patients for this analysis. The general database of patients with aortic valve diseases includes more than 2000 people. Patients for this purpose were included according to the inclusion and exclusion criteria for the period of time: 2014-2020. Due to the lack of data, less than 5% of patients were excluded.

-        A paragraph with limitations is missing. Special limitations of this analysis include missing long-term follow up, retrospective analysis.

Retrospective analysis certainly has inherent limitations. We have added this section.

As for long-term observation, this was not the purpose of this work and the long-term results will be considered by us in a separate paper.

-        What should clinicians do when inflammatory markers are elevated? Reschedule? Cancel?

Of course, the decision about the surgical intervention is decided by a HeartTeam of specialists. Systemic inflammation, like any other factor that increases the risk of developing any adverse outcome, requires various kinds of actions. Further studies are needed to establish the possible correction of systemic inflammation in specific groups of patients. For example, it may have been taking colchicine for a certain long time, both before and after surgery. At present, there are already encouraging results of studies by CALCOT, LoDoCo, etc. or other non-drug ways to suppress chronic inflammation. https://www.nature.com/articles/35013070? This requires further study.

Reviewer 4 Report

This work is about inflammation markers and TAVR. SIRI, SII, AISI, and NLR as novel biomarkers of systemic inflammation were associated with in-hospital mortality. In addition, SIRI, NLR, and MLR were associated with early postoperative bleeding. Of all markers and indices of systemic inflammation in our study, SIRI was the strongest predictor of a poor outcome in the multivariate regression model.

1. Could the author provide the data on CRP level and the correlation between novel inflammation markers?

2. Could the author add the analyses of the ROC curve of SIRI, SII, AISI, and NLR for mortality?

Author Response

We are grateful to the Reviewer for the high evaluation of our article.

  1. Unfortunately, C-reactive protein, as well as interleukins, are not included in the list of laboratory tests that are performed routinely in all patients before heart surgery, both in our clinic and in many other clinics. This is due to an additional financial burden. Therefore, we did not measure the level of CRP in these patients. The correlation cannot be estimated in this paper. Of course, it would be interesting.
  2. Although this duplicates the data shown in the Table 2, at the request of the Reviewer, we added ROC curves for mortality of all studied markers of systemic inflammation (Figures 1a-1f).

Round 2

Reviewer 2 Report

Please give my comments from the first version to a Statistician.  You  misinterpreted several of my comments.

None